# Cell Responsiveness to Physical Energies: Paving the Way to Decipher a Morphogenetic Code

**DOI:** 10.3390/ijms23063157

**Published:** 2022-03-15

**Authors:** Riccardo Tassinari, Claudia Cavallini, Elena Olivi, Federica Facchin, Valentina Taglioli, Chiara Zannini, Martina Marcuzzi, Carlo Ventura

**Affiliations:** 1ELDOR LAB, National Laboratory of Molecular Biology and Stem Cell Engineering, National Institute of Biostructures and Biosystems, CNR, Via Gobetti 101, 40129 Bologna, Italy; riccardo.tassinari@eldorlab.it (R.T.); claudia.cavallini@eldorlab.it (C.C.); elena.olivi@eldorlab.it (E.O.); valentina.taglioli@eldorlab.it (V.T.); chiara.zannini@eldorlab.it (C.Z.); 2Department of Experimental, Diagnostic and Specialty Medicine (DIMES), University of Bologna, Via Massarenti 9, 40138 Bologna, Italy; federica.facchin2@unibo.it; 3INBB, Biostructures and Biosystems National Institute, Viale Medaglie d’Oro 305, 00136 Rome, Italy; martinamarcuzzi9@gmail.com

**Keywords:** physical energies, mechanical vibration, electric fields, electromagnetic radiation, microtubuli, morphogenesis, morphogenetic code, stem cells, regeneration, cancer

## Abstract

We discuss emerging views on the complexity of signals controlling the onset of biological shapes and functions, from the nanoarchitectonics arising from supramolecular interactions, to the cellular/multicellular tissue level, and up to the unfolding of complex anatomy. We highlight the fundamental role of physical forces in cellular decisions, stressing the intriguing similarities in early morphogenesis, tissue regeneration, and oncogenic drift. Compelling evidence is presented, showing that biological patterns are strongly embedded in the vibrational nature of the physical energies that permeate the entire universe. We describe biological dynamics as informational processes at which physics and chemistry converge, with nanomechanical motions, and electromagnetic waves, including light, forming an ensemble of vibrations, acting as a sort of control software for molecular patterning. Biomolecular recognition is approached within the establishment of coherent synchronizations among signaling players, whose physical nature can be equated to oscillators tending to the coherent synchronization of their vibrational modes. Cytoskeletal elements are now emerging as senders and receivers of physical signals, “shaping” biological identity from the cellular to the tissue/organ levels. We finally discuss the perspective of exploiting the diffusive features of physical energies to afford in situ stem/somatic cell reprogramming, and tissue regeneration, without stem cell transplantation.

## 1. A Background of Questions

Increasing and compelling experimental evidence show that biology is fashioned and regulated not only by chemical signaling, but even by multifaceted physical energies, including mechanical waves, electric patterning and gradients, as well as electromagnetic radiation, which also includes light.

A major challenge that has always and long badgered scientists of multiple disciplines is trying to decipher the mechanisms underlying the causal relationship between chemical/molecular patterning at cellular level, and the timely construction and unfolding of the variegated shapes that comprise organs and tissues, up to the specification of entire individuals.

Within this context, many fundamental questions either remain unsolved, or are addressed in such a way that only scratches the surface of a largely unexplored territory.

How is information originated and fashioned from the molecular to the cellular and intercellular level, up to the point of defining the nanotopography, the micro-, and then the macro-anatomy of a given tissue/organ system?Many of the processes that have been essential to build up and complete embryogenesis do not cease after birth, but they are timely resumed throughout the entire lifespan. Are these processes essential to maintain the biological identity at the cellular- and the larger anatomical-scale level? In the affirmative, how are they intertwined with the adulthood unfolding of our biology? Alternatively, but perhaps not mutually exclusively, are the anatomy and the biology of an adult individual originated through an afterbirth chronobiology of self-organizing embryonic processes?Why are some of these embryonic-like patterns overexpressed throughout diseased, degenerative, or cancer states?How can we interpret, and deploy at the therapeutic level, the finding that cancer entails relevant embryogenetic traits, and the observation that tissue regeneration itself may drift towards tumorigenesis, using common pathways in both embryogenesis and cancer? In addition, at what/up to which level does normal coherent tissue regeneration avoid such drift, while making use of the same or similar armamentarium?What about the increasing evidence that degenerative diseases, affecting organs with highly specified anatomical diversities and functions, are nevertheless sharing an astonishing superimposable molecular misfolding? Highly diversified tissues nonetheless show a trans-organ nature in the way they are affected by similar molecular derangement(s). This applies, for instance, to the presence of similar misfolded proteins in the brains of subjects with neurodegenerative states and in patients affected with cardiac amyloidotic failure, or cardiac Tau-pathies.Overall, how are non-local, long-range communicating patterns established outside the neurally mediated connection(s)?

## 2. An Ensemble of Physical Energies Acting as the Control Software for Chemical Signaling

### 2.1. The Molecular Level: Viewing Biology at the Single-Component Feature

At a very general level, when trying to reconnect our biology to the universe in which we are immersed and are part of, we cannot avoid thinking about ourselves as a part of the vibratory nature of the universe itself. If we start thinking this way, and looking at the scientific evidence that may corroborate or neglect the approach of seeing biology with the eyes of physics, we will indeed find mounting evidence that physical forces are essential in the orchestration of living organisms. A number of interrelated processes can be deciphered with this approach, including:The physical dynamics of molecular folding;The establishment of nano-architectonics using supra-interactions;The biomolecular recognition and signal propagation afforded through the dynamics of molecular synchronization and swarming.

In all these processes, the inherent underlying mechanism at the physical level may be viewed as a form of vibration. Sophisticated tools, including atomic force microscopy (AFM) [1,2,3,4], scanning tunneling microscopy (STM) [5,6], terahertz near-field microscopy (THz-NFM) [7], and hyperspectral imaging (HSI) [8,9,10], are now helping to offer a glimpse of vibrational patterning at the subcellular and cellular levels.

A single peptide molecule can be viewed as an array of repetitive helix–loop–helix domains. These modules are intrinsically oscillatory, with the helix acting as a spring (oscillator) and the loops as inter-oscillator linkers. However, these springs are highly polarized, with arrays of positively charged amino groups (such as those in lysine and arginine) and negatively charged carboxyl groups (such as those in aspartate and glutamate). As a result, the helices in a peptide are far from behaving like a pure mechanical oscillator, but they act, rather, as an electromechanical oscillator, harboring features of interacting chemically and physically with other similar helix–loop–helix molecules. Seeing these molecules as electro-mechanical actuators implies that their mechanical oscillation is also capable of generating an electric field with radiation characteristics. Moreover, a consistent number of signaling molecules include, with their vibrational features, the capability of absorbing and emitting light within defined domain spectra, therefore being deemed “chromophores”. The list of these molecules is remarkably increasing, and includes flavins, flavoproteins, and cytochromes [11,12,13,14,15], such as those responsible for the generation of reactive oxygen species (ROS) and nitric oxide [14,16,17,18], which act as essential pleiotropic players in cellular dynamics. It is intriguing that, albeit the real extent of chromophores occurring in mammalian cells remains to be established, it is now accepted that crucial signaling paths in these cells are controlled by opsins, a group of cis-retinal dependent G-protein coupled receptors, encompassing members of the family of transient receptor potential cation channels (TRPs) [13,19,20]. TRPs span multiple members within a superfamily of molecules which are selectively modulated by defined wavelengths of light, playing a major role in cellular dynamics [21,22,23,24,25], as photoentrainment and cellular circadian rhythms [26].

It is conceivable that nanomechanical and electromagnetic oscillations will coexist as intimately connected features in these cellular actuators, since mechanical waves will hardly separate from the generation of electric oscillations and electromagnetic radiations. Conversely, the exposure of these cellular actuators (including signaling peptide molecules) to an electromagnetic field or light will produce mechanical oscillations [5]. Compounding the complexity of this scenario is the evidence that the effects elicited either by mechanical or electromagnetic (including light) energies are remarkably more accentuated, and recruit a wider repertoire of cellular responses and fate decisions, when these physical energies are delivered as pulse-modulated rather than continuous waves [27,28,29,30,31].

These observations strongly suggest that:The physical properties of signaling molecules intrinsically embed mechanical and electromagnetic patterning, as a whole, in an inseparable nanoworld of physical forces;As a consequence, we prefer referring to vibrations as an “inclusive term”, symbolizing the oneness in the way the same forces may emerge from, and be sensed by, cellular molecules;Such physical characteristics are fashioned to be expressed as, and consequently, respond to, precise signatures.

The function of cellular proteins is tightly related to their dynamic flexibility and specific structural changes. These changes have long been known to be essential in the establishment of cellular communication and fate decisions. Nevertheless, only recently, owing to the use of THz-NFM, has it been possible to infer that flexibility and structural changes occur as a precise pattern of vibrations, and that such vibrations are organized as (i) long-range modes, (ii) under-damped modes, existing for frequencies greater than 10 cm^−1^, and (iii) persisting motions within the under-damped modes. The occurrence of these persisting motions indicates that damping and intermode coupling are both weaker than previously assumed, and suggest that, like violin strings or pipes of an organ, cellular molecules can vibrate in different patterns within our cells [7].

A logical consequence of these findings is also that the signaling features of molecules involve the capability to both express and respond to precise vibrational signatures.

Consonant with this expectation, innovative approaches have been developed that are based upon the exploitation of vibrational features of molecules to afford further characterization of their properties under both physiological and diseased states. One of these approaches is the resonant recognition model (RRM) [32]. The RRM is a method that treats the sequence of proteins and other signaling molecules, including DNA, as a precise signal, based upon evidence that defined vibrational frequencies (periodicities) in this signal characterize the biological function of molecules [32,33]. RRM is based upon the finding that protein function may be controlled by periodic distribution in the energy of their delocalized electrons, thereby modulating protein–protein interplay, as well as protein–DNA interactions, a fundamental step in DNA remodeling and the epigenetic control of biological properties in living organisms. Viewing vibration of molecules as an inseparable convergence of mechanical and electromagnetic forces, the RRM also postulated that protein conductivity could be associated with defined spectral signatures, resulting from electromagnetic radiation/absorption patterns generated by the flow of electric charges through the protein backbone [34,35]. In particular, the RRM proposes that not only the characteristics of single molecules, but also the molecular interactions that control the biological processes rely upon electromagnetic resonances between interacting biomolecules at specific electromagnetic frequencies within the infra-red, visible, and ultraviolet frequency ranges. Each interaction can be identified by the defined critical frequencies intervening in the resonant activation of targeted functions in proteins and DNA.

Another advantage of the use of RRM is the chance of developing novel peptides with unprecedented spectral features and bioactivities [36].

Interestingly, the spectral signatures predicted by RRM for resonance frequencies of tubulin have been verified and supported by experimental evidence [5,35]. In fact, STM, coupled with an artificial cell replica developed to deliver electromagnetic fields of specific frequencies to tubulin molecules assembling onto platinum nanoelectrodes, has shown that tubulins, tubulin dimers, and microtubules exhibited electric conductivity profiles resonating only with specific electromagnetic frequencies applied through the cell replica system [5]. STM analysis also provided evidence that the resonant tunneling currents elicited by microtubules occurred in response to electromagnetic fields applied within a MHz range [5]. These findings indicate that a single tubulin molecule can generate specific electromechanical oscillations as a consequence of a resonant response to defined electromagnetic frequencies produced or delivered within their environment [5]. The same conclusion applies to the assembled structures, such as tubulin dimers and the more complex microtubuli (Figure 1), which provide a vivid example of the creation of complex nano- and micro-architectonics using supramolecular interactions. Even more intriguingly, these findings further support the notion that mechanical and electromagnetic resonance modes can coexist and affect each other within the same molecular network (as it has been shown for tubulins and microtubuli).

A major challenge in the biophysical interpretation of cellular dynamics is moving from the analysis of single-molecule vibrations to the dissection of complex collective behaviors on a vibrational basis. How can biomolecular recognition patterning within an interactome be figured in terms of multiple interacting vibrational signatures?

### 2.2. From the Single-Molecule Level to the Collective Behavior: Approaching Biomolecular Recognition

Answering these questions has been made possible by the increasing understanding of the physics behind the structure and function of the cytoskeleton, with particular emphasis on cellular microtubuli. It is now evident that these elements form a complex vibrational network involving the generation of concerted electromechanical waves, where mechanical vibrations (Figure 2) and mechano-sensing activities are intimately connected with the generation of electric and electromagnetic oscillations, imparting features characteristic of connectedness and long-range force radiation. In a bottom-up approach, single microtubuli and their small networks interconnected in vitro as linear and triangular geometries became remarkably stiffer on short time-scales upon mechanical stimulation [37]. In particular, a substantial stiffening of single microtubuli was observed above the defined transition frequencies of 1–30 Hz, depending on the microtubular molecular composition. Below this frequency, microtubule elasticity only resulted from its contours and persistent length. Such elastic behavior could be transferred to small networks, where linear two-filament connections acted as transistor-like, angle-dependent momentum filters, whereas triangular networks behaved as stabilizing elements [37]. On the whole, these findings revealed that microtubuli, and likely, intact cells, may tune mechanical patterns through unexpected temporal and spatial filter capabilities. These observations prompt further studies aiming at elucidating the functional role of microtubuli in different cell types on the basis of the complexity of their structural assembly and their networking with other cytoskeletal associated components.

In addition to their mechanical properties, microtubuli have been found to exhibit sophisticated electric signaling features. The possibility that these elements emit high-frequency electric fields with radiation characteristics had been initially postulated on theoretical basis by Havelka and coworkers [38]. Subsequently, by the aid of a microelectrode system, Santelices et al. recorded small-signal alternating current (AC) conductance of electrolytic solutions containing microtubuli and tubulins [39]. These authors found that, in electrolyte solutions, microtubuli increased conductance at 100 kHz in a manner dependent upon their concentration, while a concentration-independent peak in conductance was detected at 111 kHz, which appeared to ensue from an intrinsic property of microtubuli in an aqueous microenvironment [39]. Intriguingly, tubulin dimers were found to decrease solution conductance under the same experimental conditions. This finding indicates that the self-assembling of microtubuli results in the formation of nanowires with defined electrical signaling, and that the capability to modulate the conductance of aqueous electrolytes may have profound implications for cellular bioelectrical processing and communication. This hypothesis is further corroborated by the evidence for electrical oscillations in two-dimensional microtubular structures [40]. This configuration enhances the experimental accessibility of the interspersed nanopores formed by the lateral subunit arrangement in the structural wall of the microtubule. Voltage-clamped microtubular sheets produced cation-selective electrical currents of a magnitude tuned by ionic strength and composition, as well as by the holding potential [40]. Moreover, current injection generated voltage oscillations, indicating an action potential-like excitability. The documented genesis of electric oscillations in microtubuli may pave the way for a deeper understanding of the non-linear behavior of the cytoskeleton. These discoveries hold promise for remarkable therapeutic applications. In fact, exposure to alternating electric fields, ranging between 100 to 300 kHz, a range overlapping the conductance of microtubuli in electrolyte solutions [39], with strength of about 1–2.5 V/cm, has been found to disrupt cancer cell replication [41] and arrest cell proliferation in vitro in animal tumor models, and even in human brain tumors, within the course of a pilot study in patients with glioblastoma multiforme (GBM) [42]. These discoveries have led to a Food and Drug Administration (FDA)-approved treatment of GBM using tumor-treating electric fields (TTfields) in addition to standard treatments [43,44]. In the final analysis of this large randomized clinical trial, involving 695 patients treated with standard radio-chemo-therapy, the addition of TTfields to the maintenance temozolamide chemotherapy resulted in statistically significant improvement in progression-free survival and overall survival, as compared to the treatment with temozolamide alone [44]. These exciting developments prompt additional investigations, aiming at a deeper elucidation of the mechanistic bases of microtubule dynamics in the hope to further refine and improve the outcome of these clinical applications. As an initiative that may prove rewarding in this direction, very recently, Havelka and coworkers have developed a lab-on-chip microscope platform allowing for the investigation of the effects of microsecond pulsed electric fields (PEF) on microtubule networks of cell-like density [45]. They found that PEF stimulation resulted in an aggregation of microtubules, overcoming the non-covalent microtubule bonding force to the substrate. These results indicate that establishing bioelectronic circuits at the microtubular level remarkably affects their mechanosensing and nanomotion properties, and hold promise for future developments of bioelectronics technologies and electromagnetic tools for the manipulation of structural and signaling properties of the cellular cytoskeleton.

Compelling evidence for previously unexpected features of microtubuli came from the discovery that they even entail multi-level memory-switching properties. By the aid of sophisticated STM analyses, the formation of these memory states has been found to involve vibrational (mechanical and electromagnetic) modes that control protein arrangement symmetry related to the conducting state written within the nanowire structure of the microtubule. These features were found to enable a single microtubule to store and process about 500 distinct bits, with 2-pA resolution, creating a random access memory analogue of the flash memory switch used in a computer chip [46].

Microtubuli have attracted even more interest from the physicist community, as their structures share some similarities with photosynthetic antenna complexes, particularly in the ordered arrangement of photoactive molecules with large transition dipole moments. The analysis of tryptophan molecules, the amino acid building block of microtubules with the largest transition dipole strength, has been conducted by taking their positions and dipole orientations from realistic models capable of reproducing tubulin experimental spectra, and using a Hamiltonian widely employed in quantum optics to describe light–matter interactions. This approach showed that such molecules, arranged in their native microtubule configuration, exhibit a superradiant lowest exciton state, which represents an excitation fully extended on the chromophore lattice [47]. Theoretical insights have also been provided that such a superradiant state emerges due to supertransfer coupling between the lowest exciton states of smaller blocks of the microtubule. In this modeling, the velocity of photoexcitation spreading was shown to be enhanced by the supertransfer effect, with respect to the velocity one would expect from the strength of the nearest-neighbor coupling between tryptophan molecules in the microtubule [47]. Additionally, these structural configurations exhibited enhanced resistance to static disorder, in comparison to short-range interacting geometries that include only short-range interactions. The possibility that this superradiance and supertransfer cooperative behavior may be at the basis of ultra-efficient photoexcitation absorption, and could enhance excitonic energy transfer in microtubules over long distances under physiological conditions, forms an interesting theoretical premise for subsequent experimental challenges.

An overall picture emerges where the microtubular network, and more generally, the cytoskeleton, can be regarded as a complex “bioelectronic circuit”, capable of generating a continuous flow of information fashioned as a multitude of self-organizing vibrations, emerging and diffusing as short- and long-range mechanical/electromagnetic (including light) oscillatory patterns. This bioelectronic circuit is also the context where biomolecular recognition and further information is generated. In fact, the extremely high speed and coordinated tuning of molecular interplay at the basis of complex cellular decisions, including the orchestration of stem cell fate, cannot be solely explained by the diffusion and collision of molecules within the aqueous intracellular environment. A number of interrelated observations argue against a diffusive model in the scaling-up of information by interacting molecules: (i) the establishment of free-diffusion trajectories will be minutely disturbed by the continuous motion of cytoskeletal elements; (ii) the cellular interior is continuously remodeled by motions of a great number of subcellular organelles; (iii) continuous fluid perturbations are generated by a hive of nanovesicles (exosomes) moving along cytoskeletal tracks from their multiple sites of fabrication and “refinement” up to the cell surface, where they are released to contribute paracrine and autocrine signaling; (iv) the microtubular network is also the tracking system for highly polarized motions of molecular motors, which will increase the density of intracellular trafficking and hamper diffusive mechanisms; (v) the formation of tunneling nanotubes, now emerging as an additional tool of physical intercellular communication, involves remarkable contribution by cytoskeletal elements and further traffic crowding; and (vi) the synthesis and accumulation of a wide variety of intracellular glycosaminoglycans, such as hyaluronans, impart the features of a non-homogeneous aqueous gel, dynamically modifying its composition and diffusive properties in response to cell metabolism.

Taking into account these considerations, a biomolecular recognition based upon diffusive mechanisms and lock-and-key interactions among molecules appears to be highly unlikely, entrusted to an extreme degree of randomness and unpredictability.

A reliable alternative view of biomolecular recognition and its role in the emergence of biological information may result from translating the vibrational properties of individual molecules/oscillators into the collective behavior of multiple oscillators placed and moving within the context of a visco-elastic microtubular network. Signaling proteins can be equated to a crowd of oscillators using molecular motors to move along the microtubular net.

An important characteristic in microtubule networks is the establishment of synchronization patterns, as well as the exhibition of a collective behavior. Synchronization is a self-organization modality which manifests in a wide variety of natural as well as manufactured technological systems, ranging from excitable cells (such as neural or pacemaker cells) to coupled lasers, metallic rods, and robots. The finding that simple mixtures of microtubules, kinesin clusters, and a bundling agent assemble into spontaneously oscillating structures suggests that self-organized vibration modes may emerge as a generic feature of internally driven bundles [48]. Synthetic cilia-like elements self-assemble at high density, exhibiting synchronization and metachronal traveling waves, recalling those observed in natural ciliary fields [48]. Microtubuli have exploited their synchronization properties, evolving from elements that provide shape, form, motility, anchorage, and apparatuses for feeding in simple protists to structures that control cell polarity and cell signaling networks in complex eukaryotes.

Another form of self-organization is the swarming of insects, flocking of birds, or schooling of fish, where a collective behavior emerges as the movement of a multitude of individuals through space without significant change in their internal state(s) [49]. Pioneering a new field of enquiry, Sumino and coworkers provided the first evidence that an artificial network of microtubules propelled by molecular motors (dyneins) self-organized into a pattern of whirling rings [50]. These authors discovered that colliding microtubules align with each other with high probability. As a function of increasing microtubular density, their alignment resulted in a self-organization into vortices of defined diameters, inside which microtubules were observed to move in both clockwise and anticlockwise fashion [50]. This collective behavior not only occurred as a spatial arrangement, but the phenomenon also exhibited a timely unfolding, since, over time, the vortices coalesced into a lattice structure. The emergence of these structures appeared to be the result of smooth, reptation-like motions of single microtubules in combination with local interactions (collision-dependent nematic alignment) [50].

These discoveries have brought to the fore a previously unsuspected universality of synchronization and swarming, as the modality through which individual oscillators may converge into a collective behavioral network.

Increasing experimental evidence is now indicating that signaling molecules may not only exploit their individual function, but even recognize each other and establish informational networks through specific vibrational patterning. Structural vibrations steer protein structure to functional intermediate states, as has been shown by anisotropic terahertz microscopy and simplified versions of this technique, revealing specific vibrational protein signatures with remarkable functional implications [51]. These approaches are now providing unique access to the directionality of intramolecular vibrations and their functional meaning. The biological relevance of vibrational direction versus the energy distribution has been highlighted when using these measurements to compare wild-type lysozymes with a double-deletion mutant with a higher catalytic rate [52]. While no difference in vibrational distribution was detected, a significant re-orientation of vibrational modes could be demonstrated for the more efficient mutant [52]. Robust modes of ensemble analysis have allowed for the calculation of persistent protein motions, as the narrow frequency bands corresponding to the vibrations of hen egg white lysozymes overlap with its simulated functional dynamics [53]. An analysis of temperature-dependent activation energy in lysozyme and cytochrome c, using terahertz permittivity measurements, revealed the functional-state dependence of picosecond protein dynamics, with the presence of both higher and lower activation energy states—the latter being consistent with collective structural motions and disappearing upon protein denaturation [54]. The functional relevance of these observations was further corroborated by terahertz time-domain spectroscopy and molecular dynamics simulation, revealing structural collective protein motions on a picosecond timescale of heme protein, and cytochrome c as a function of oxidation and hydration [55]. Recent observations, using femtosecond optical Kerr-effect spectroscopy, have shown the presence of low-frequency vibrational modes in G-quadruplexes and in B-DNA, characterized by underdamped delocalized phonon-like modes with the potential to even modulate the biology of DNA at the atomic level [56]. Cumulatively, these observations reveal the chance of detecting the presence of incredibly fast, precise, tunable, and directionally oriented vibrational signatures as a major determinant in the steering function of protein. Nevertheless, these data are still pointing at the relevance of vibrational patterns in determining the function of individual proteins or signaling molecules in general.

An analysis of collective motions in the DNA-binding domain (DBD) of wild-type, mutant, and rescued mutant forms of p53 indicated that similar defined low-frequency vibration patterns could be detected in the wild-type and rescued mutants [57]. The mutations in the DBD affected the low-frequency vibration of the p53 tetramer by modifying the collective vibrations among its four monomers [57]. These findings help to establish a correlation between vibrational patterns in signaling proteins and biomolecular recognition, an essential step in the onset and propagation of informational processes in living organisms.

On this basis, signaling peptides can be regarded as a multitude of oscillatory devices using molecular machines (such as kinesin motor proteins) to move along the microtubular net, with the microtubules acting themselves as multi-level connectors affording efficient phase synchronization between multiple oscillators. Although coherent synchronization and chaotic behavior have long been deemed as mutually exclusive in a network of similar/identical oscillators, there is now evidence that chimeric states may spontaneously ensue from contention among antagonistic synchronization patterning within oscillators joining a hierarchical web [58].

Martens et al. have devised non-local coupling in a hierarchical network encompassing two subsets of oscillators, with each subset embedding strongly coupled oscillators, but with a weaker coupling strength between the two subsets. Identical metronomes with a given oscillatory frequency (*f*) were placed on two swings, freely moving in a plane. Within one subset, the oscillators became strongly coupled by the motion of the swing they were bound to. Increases in *f* resulted in increased momentum transfer to the swing, which led to stronger coupling between oscillators (metronomes). Upon the increase of the coupling within the same subset, a single swing underwent a transition phase, from a chaotic to a coherent (synchronized) state. In this system, self-organization was controlled by elementary ubiquitous mechanics found in many natural and artificial technological systems [58]. These findings show that break-of-symmetry dynamics may represent a remarkable trait within networks manifesting a collective behavior, irrespective of whether the network lays in a power grid, optomechanical devices, or a living cell.

Inside the cell, signaling molecules recognize each other, creating a network of information which orchestrates essential processes, including growth, differentiation, survival, death, or aging.

Within the context discussed herein, microtubuli may be considered as an elastic matrix of nanowires capable of mechanical and electromagnetic oscillatory patterns with radiation characteristics, while, taking into account the previously discussed RRM [32,33,34,35], signaling molecules may be equated to oscillators, behaving as resonators for frequencies in the range of 10^13^–10^15^ Hz, consonant with the length of amino acid chains. With an inter-amino acid distance of about 3.8 Ä, the estimated velocity of electric charge along the protein chain has been estimated around 7.87 × 10^5^ m/s [32,33], which led to considerations of the establishment of protein-mediated resonance frequency ranges with radiation characteristics. Thus, the resonance frequencies of proteins and signaling molecules, sustained by their inherent vibrational patterns, may be, therefore, exploited as synchronization modalities within clusters of oscillatory resonators. According to the above-reported experimental findings of Martens and coworkers [58], the microtubular network may act as a system dissipating vibrational differences and facilitating synchronization modes between signaling oscillators bearing similar vibrational frequencies. We may also speculate that, while promoting/facilitating synchronization between signaling players, the microtubular network will undergo remarkable changes in its own mechanical and electromagnetic dynamics, further contributing to the manifold electric and electromagnetic phenomena occurring at the intra- and inter-cellular level. In this complex scenario, further dynamicity and complexity is added by the role of molecular motors, which act as remarkable conductors in the transport of multifaceted cargoes, including signaling peptides, within the crowded, viscous environment of living cells [59]. Experimental data now show that submicron particles, and even nano-scaled signaling molecules, which can hardly and anomalously diffuse within the viscoelastic cellular cytosol, can be subjected to fast directional transport only thanks to the action of molecular motors [60,61]. To this end, the mechanochemical coupling between molecular motors, their cargoes, and the cytoskeleton, as well as the networking dynamics of multiple motors and cargoes, play a fundamental role, and their dissection is becoming a major field of enquiry [62,63,64]. Notably, the analysis of the physical crowding of cargo movements inside the cells may contribute a deeper understanding of diseased states [64,65]. Further studies are, therefore, desirable to elucidate the role played by molecular motor trafficking in biomolecular recognition, within the frame of the currently discussed electromechanical and frequency resonance patterns of cytoskeletal elements and signaling molecules.

Considering the unavoidable incompleteness of the current knowledge, we may take advantage of a metaphoric representation to help summarize the complexity of this matter. We may imagine an acrobat (the molecular motor), carrying on her shoulder some precious cargo full of information (the signaling molecule), gently walking on an elastic wire (the microtubule). On a collective scale, we may think of an ensemble of dancers moving through a network of these nanowires. Recall the metronome experiments of Martens and coworkers [58]: each signaling molecule (oscillator), per se, is like a metronome, and the wires (microtubuli) are facilitating the attainment of coherent synchronized states among these metronomes.

This means that it is easier for these metronomes to achieve a synchronous state, and while they will remain individual, each element of the network will become aware of what is occurring in the system as a whole because of its interconnectivity. This entire approach uses tenets from nanomechanics, electromagnetism, and quantum field theory, since it aims at taking a glimpse at how multiple patterns can share information. We are becoming aware that microtubuli within the cell behave as migrating oscillatory tracks, and that their “dancing” is associated with a remarkable generation of electric fields, spreading information across the cells, and trespassing the cell boundaries. Further dissection and understanding of these dynamics will very likely provide novel cues for reinterpreting cell-to-cell communication.

Like a symphony, in the cellular orchestra we may envision multiple coherent domains, each with a population of oscillators synchronized on specific frequencies, waveforms, and/or pause intervals. An additional level of self-organization may be achieved through the swarming of multiple coherent domains, each retaining their internal state of synchronization while migrating across the cell.

### 2.3. Electromagnetic and Nanomechanical Energies in the Modulation of Stem Cell Biology

The capability of cells to act as both senders and receivers of physical signaling discloses the chance to exploit completely new actions at biomedical level. One possibility is to reprogram stem cells and enhance their rescuing potential without relying upon the use of cumbersome chemistry, or viral vector-mediated gene transfer, an approach which is still not readily envisionable in clinical practice. The strategies of modulating stem cell biology with electromagnetic fields [66,67,68,69,70,71,72], even reversing stem cell senescence in vitro [73,74], or driving cell fate mechanically by shock waves [75], as well as the possibility to use AFM to harvest and release precise vibrational signatures, affording the fine-tuning of stem cell dynamics [76], have long been a focus of our studies.

Comprehensive review analyses, focusing on the rescue of damaged tissues by the aid of physical energies, also including the use of shock waves and photobiomodulation, can be found in [77,78,79].

The diffusive features of electromagnetic and mechanical waves would allow the efficient reprogramming and modulation of stem cell biology in places where the stem cells already reside—in all tissues of the human body. We believe that targeting stem cells with a physical stimulation in vivo will pave the way for an unprecedented approach in regenerative medicine, avoiding cell or tissue transplantation, and providing tissue regeneration by directly boosting the multilineage and paracrine repertoire of tissue-resident stem cells. This will basically lead to enhancing our inherent self-healing potential. The recent discovery of multilineage-differentiating stress-enduring (MUSE) cells, an SSEA-3(+) pluripotent, tissue-resident stem cell population within human mesenchymal stem cells (hMSCs) [80,81,82], may offer further opportunity to exploit direct in situ reprogramming through ad hoc conveyed physical energies. Exploring whether MUSE cells may represent an effective target for electromagnetic fields and mechanical vibrations may be of particular relevance, given the capability of these cells to differentiate into endodermal, ectodermal, and mesodermal phenotypes, and to contribute to tissue repair at serious damage sites, as shown after acute myocardial infarction [83], even in humans [84,85], and in animal models of stroke [86,87]. It is usually believed that tissue-resident somatic cells are not involved in tissue regeneration, but they can, rather, be responsible for pathological tissue remodeling, as occurs in scar formation. Contradicting such a viewpoint, we provided evidence that asymmetrically conveyed radioelectric fields are able to directly reprogram human somatic cells into cardiac-, neural- and skeletal muscle-like cells [70] in vitro. This finding suggests that the delivery of precise physical signals to human tissues in vivo may become a future strategy for multitargeting their intrinsic rescuing potential, not only by intercepting a variegated population of stem cells (including hMSCs and MUSE cells), but even by acting at the level of the resident somatic counterpart.

### 2.4. From the Cellular to the Tissue Level

Considering what we have discussed above, cells may be equated to a fractal component of a living organism, being “traversed” by mechanical and electromagnetic vibrations that self-organize in both local and non-local dynamics, similar to what occurs in the universe we are a part of. Within each cell, the cytoskeleton may be considered as a bioelectronic circuit capable of sending and receiving oscillatory patterns in the form of mechanical and electromagnetic radiation. The local and non-local transmission of oscillations and traveling waves, the assembly of stable and migrating intracellular domains, the homeostatic control in bioelectric circuitries, and the role of electromagnetic fields in the control of gene expression and cell fate [66,67,68,69,70,71,72,73,74] are now posing the needs for the development of ad hoc computational analyses to decipher how these physical forces may contribute to generating and spreading information, up to the development of morphogenetic outcomes, involving tissue, organ, and large-scale anatomy. These issues are now the subject of intensive experimental activities and are forming the platform for novel hypotheses, seeing our cells in terms of informational stations using mechanical and electromagnetic energy as a form of extremely fast and integrated connectedness to tell the biochemical machinery how to work [65,77,78,79]. This is exactly what computer software does, acting as a collection of data or computer instructions to define the working dynamics of the computer itself.

Any tissue/organ is a highly integrated multi-cellular ensemble where different cell types merge to create defined architectures that, at the embryo level, proceed in the hierarchical definition of the cranial–caudal axis. Intriguingly, tissue regeneration recapitulates multiple hallmarks of embryonic development, repurposing lineage-specific profiles to re-initiate morphogenesis and tissue rescue [88]. This observation suggests that, in adulthood, a part of embryogenetic mechanisms can be resumed to maintain our biological identity and provide remarkable biophysical flexibility, adaptation, and short- and long-term biological memory. This implies developing an increased number of functions, a process which entails the capability of evolving. Cells are, therefore, embedding, self-organizing chemical structures, such as microtubuli and other cytoskeletal elements, that now appear to be “written” by a software encompassing physical energies of an electromagnetic and mechanical nature. In this frame, we may refer to the genome, the proteome, and other omics (such as the kinome) as the hardware of our biological systems [89,90].

While, in referring to the structure–function relationship, we often think in terms of multi-component/multi-task target functions, biology has brought us the notion of one-component/multiple-task target functions. As an example of the first category, we may include the seven helices G-protein-coupled receptors, where multiple components (the seven helices, their interspacing loops, and trimeric G-proteins) afford multiple tasks and interact with multiple targets in the related signaling cascades. An example of the second category is provided by tyrosine kinase receptors, or serine-threonine kinase receptors, with their monomeric G-proteins exemplifying a system in which one or a few elements control cascades of multiple protein kinases and, therefore, their related downstream signaling. Other such examples include many so-called pleiotropic agents that act as a single-component, affecting the structure and function of a wide variety of signaling players. To this end, DNA has been reported to have remarkable characteristics of an electromagnetic fractal cavity resonator, which can interact with the electromagnetic fields over a wide range of frequencies [91].

Using the lattice details of human DNA, the radiation of DNA has been modeled as a helical antenna, with the DNA structure resonating with electromagnetic waves at 34 GHz, with a positive gain of 1.7 dBi [91]. Based upon the analysis of the role of three different lattice symmetries of DNA, the possibility of soliton-based energy transmission along the structure has also been formulated [91]. If confirmed by subsequent investigations, these findings may place DNA at a point of convergence where the chemical structure of its gene sequences will serve as a multi-component/multi-task target moiety, while its physical behavior of a helical antenna will make the same molecule working as a one-component/multiple-task target entity capable of sensing and transducing electromagnetic waves of different frequencies and modulation characteristics. Compounding a chemical/physical assessment of nuclear material, it has also been proposed that the density and positioning of nucleosomes in a chromosome region, the physical properties of histone octamers, and the interplay of chromatin-binding complexes within a chromatin domain will cooperate to define the physical strength of that domain [92]. Multiple chromatin domains are now shown to physically oscillate, manifesting the ability to form clusters among domains that share similar oscillation frequencies. Such synchronized clusters are believed to function as chromatin organizers involved in shaping higher-ordered chromatin structures, and associated epigenetic regulations [92]. In this way, software and hardware dynamics coexist in space and time, and may spread in—and out—of the nucleus, thanks to complex array of pleiotropic molecules “affiliated” with the LINC (the linker between the nucleoskeleton and cytoskeleton). In this regard, the LINC may provide synchronization and swarming between multiple oscillators inside and outside the nucleus. Local vibrational changes at nuclear or other subcellular levels would afford a timely modulation of the actual oscillatory patterns inside the cytoskeleton [93,94,95]. The presence of adhesion molecules, gap junctions, and tunneling nanotubes, may, then, afford the propagation of vibrational modes among neighboring cells.

While the spreading of mechanical oscillations in the cytoskeleton may be assumed as a relatively short-range spreading phenomenon, the diffusion of electromagnetic (light included) waves through the cytoskeletal elements may be envisioned as a long-range disseminating phenomenon, theoretically circulating through an entire organism.

Considering cell biology in the light of software- and hardware-like dynamics may provide deeper understanding and control of developmental as well as regenerative pathways, and may place diseased states, such as degenerative diseases and cancer, within the context of an unplugging of the cellular hardware from the network of physical energies organized as a cellular software.

With the advancement of technologies, and in particular with the progress in the field of optics, understood as the ability to interact with the dual, mass/wave nature of light, research studies, even in the biological field, have launched into a new era that is not determined by chemistry alone. There is more and more evidence, at first only corroborated by observation and intuition, that there are phenomena, which we now know are related to quantum physics, that are capable of governing the biological and biochemical mechanisms of cells in particular, and whole individuals in general. In nature, quantum mechanisms are used in practically all fields in which matter must interact with light: photosynthesis, for example, exploits the now well-studied phenomena of tunneling and entanglement to produce energy, and manages to do so in what was once thought to be a “forbidden” field to quantum effects, that is, a living and hot system [96,97,98]. In fact, the idea that quantum phenomena can exist only at low temperatures and in low-entropy environments has been completely denied by the last fifty years of research. Already from the birth of the same idea of the atom, many physicists imagined that multiple rules could also apply to living and vital systems, and their intuitions have been confirmed today. Niels Bohr, one of the fathers of the atom, was the author of a lesson called “Light and Life” [99], which already laid the foundations for the existence of a wave (mass or light)–biology interaction. Max Delbrück, Nobel laureate in Physiology or Medicine, despite not having the proper instrumental tools at his time, already ruled out that biological life was structured only on the basis of mere chemical processes [100]. In his 1944 treatise “What is life?” [101], Erwin Schrödinger theoretically postulated the existence of a “crystal containing information” when DNA was not even known, stating that life could not be limited to classical chemistry and physics, which were not and would not be able to give the most intimate answers for the concept of life itself. Only 60 years later, instruments capable of probing light itself have proven these apparently visionary and almost spiritual theories to be right.

Among the main players in this field, we find cryptochromes, proteins for which the name itself suggests the complex and mysterious nature of their action. It has been discovered that these proteins, involved in essentially all the interaction processes with electromagnetic waves, are fundamental for collecting the signal and translating it into a classic “voltage-dependent” or “chemical-dependent” pathway [12,102,103,104,105]. They act as a bridge between the quantum world and structures above nanometers that are the cellular ones, and by their nature, they are able to work at speeds unthinkable for chemistry. As a vivid example, the cryptochrome Cry2 is remarkably expressed in the human retina, and it has been shown to function in a light-dependent manner as a magnetosensor in the magnetoreceptor system of the Drosophila. Our eyes, therefore, hide a molecule capable of working as a light-sensitive sensor of the earth’s electromagnetic field, even without our brain making this information available to us consciously [106]. These findings make our retina an intriguing territory for further exploring quantum biology [107,108].

Even more intriguingly, light-sensitive melanopsin neurons are located in the retina, and are tightly involved in the regulation of cognitive and mood functions in response to defined light patterning in mice [109]. Moreover, these proteins also exist in cells deeply immersed in the human body, and are able to interact with particular light frequencies. Opn4 is a non-image-forming opsin involved in circadian photo-entrainment and affective responses. Its discovery in blood vessels is, really, game-changing. The finding that vascular Opn4 mediates light-dependent vascular relaxation in a wavelength-specific fashion, opens a completely novel scenario in the study of light-mediated patterning, and in the effects elicited by photobiomodulation in biological systems [110]. So, why should a complex but intelligent system, such as a cell, use resources to express such molecular complexes? Gradually, however, it has been understood that not only particular proteins are able to react to electromagnetic fields: the DNA itself may be influenced by magnetic fields [32,33,91], and some molecules have intriguing abilities to absorb light and emit light, qualities that, in nature, are rarely delegated to case. Some issues are still very difficult to investigate. The very nature of the electromagnetic pulse is too small to be deciphered for the current measurement systems, which would modify the signal in the act of observation. Nevertheless, the final biological effects can still be investigated. Whether they are visible frequencies, such as pulsed or infrared light, particular colors, or more generally low-, or high-frequency electromagnetic waves, today, we know that cells are able to sense and read this information, and to translate it into an effect.

## 3. Bioelectricity in Living Organisms: The Emerging of a Morphogenetic Field

### 3.1. From Pioneering Studies to the New Course

The term bioelectricity points at the capability of living cells, tissues, and organisms to endogenously generate electric fields, with the potential to affect biological/functional dynamics. This field of enquiry grew progressively and has spread since the first studies, published in 1791 by Luigi Galvani in “De Viribus Electricitatis in motu musculari”, and Galvani’s subsequent discovery that a twitch can be elicited by placing a muscle in contact with a deviating cut sciatic nerve without the supply of metal electricity [111,112,113,114]. A fundamental advancement in the field of bioelectricity was achieved by the work of Emil du Bois-Remond, who demonstrated macroscopic electricity in frogs, fish, and human tissues, thus discovering the action potentials [115,116], and conclusively demonstrating the injury potential and current [117], for which Galvani himself had previously unknowingly provided evidence [114].

Following these pioneering studies, a fundamental breakthrough in the history of bioelectricity was marked by the relentless work of Harold Saxton Burr in the early 20th century (from 1916 up to the late 1950s). His studies were published in extremely relevant journals, including the Proceedings of National Academy of Sciences USA and Science. Burr developed an accurate millivoltmeter [118], and was able to trace and characterize the field properties of a developing frog’s egg [119]. In these studies, Burr performed some six thousand determinations on fifty frogs’ eggs, prior to the development of the primary axis of the embryo, as seen in the appearance of the medullary plate, recording potential differences in the electric pattern between the vertex of the terminal pole and four equidistant points on the equator of the egg [119]. The characterization of the electric field properties along the embryo development provided the first evidence that the primary axis of the organism came to lie in the plane of the greatest voltage drop from the vertex. In other worlds, Burr was able to predict, from the voltage pattern, where the head of the organism would develop, coming to the conclusion “that the electric pattern is primary and in some measure at least determines the morphological pattern” [119]. Burr also conducted studies on the electrodynamic patterns in a wide variety of plants, spanning from the growth correlates of electromotive forces in maize seeds [120] to the effect of a severe storm on the electric properties of a tree and the earth [121]. His rigorous methods, coupled with a visionary and eclectic personality, made Dr. Burr conceive that all living forms rely upon the existence of electrodynamic fields [122]. In this study, Burr declared his intention of “searching for the explanation of the phenomena, not in the currents alone but also in the surrounding medium”, drawing his attention to the field physics, rather than to the particle physics, being aware of the fact that “field physics centers theory and experimentation upon the medium in which the system as a whole is embedded and upon its structure” [122]. For these purposes, Burr designed a “vacuum-tube microvoltmeter” with a high degree of sensitivity and stability. This tool allowed him to explore the electric properties of a wide variety of living forms, with contacts between the instrument and the living organism made through silver–silver chloride electrodes immersed in physiological salt solution, with the entire apparatus being shielded and grounded at appropriate points, so that the recorded deflections of the galvanometer would have provided an accurate picture of the voltage differences in the explored living system. Remarkably, in his experiments, Burr showed that voltage gradients between the head and tail of Amblystoma or chick embryos could be determined with considerable certainty, not only when contact was made directly with the organism’s surface, but even when the electrodes were up to 2 mm away from the embryo surface [122]. With the same technique, Burr could show that the salamander embryo, revolving between the tips of a pair of capillary electrodes as a result of ciliary action, produced defined oscillations in the galvanometer as the developing head passed in a sequence under electrode pair of the system [122]. These findings gave the first evidence that, under the explored conditions, the embryo was acting as an AC generator of very-low frequencies, a phenomenon that could only be explained on the assumption that an electric field was existing and acting in the embryo. In his studies, Burr provided seminal discoveries, ranging from the first dissection of the response of slime mold to electric stimuli [123], to the discovery of defined bioelectric patterns during human ovulation [124]. Through the cooperation of a patient subjected to a laparotomy, Burr was able to perform a continuous recording of voltage differences intervening between the symphysis pubis and the vagina for 57 h, showing the feasibility of using bioelectric field assessment to determine with certainty and accuracy the time of ovulation in an intact human being [124]. Within this context, Dr. Burr also dissected the electrical signatures emerging from human diseased states, such as the electric correlates from nerve injury [125,126]. Dr. Burr addressed the electric features of cancer-susceptible mice to explore whether changes in voltage measurements may occur during the onset and development of a malignant tissue [127,128]. The results of the experiment consistently showed that twenty-four to twenty-eight hours after tumor implantation, changes were observed in the voltage gradients. This differential increased steadily and quite smoothly to reach a maximum of approximately five millivolts on or about the eleventh day. The analysis of bioelectric fields in the course of malignancies was also extended to human beings. In collaboration with Dr. Luis Langman, the approach of recording voltage gradients between the symphysis pubis and the vagina was exploited to assess whether marked changes in these gradients may reveal an early onset of malignancies [129,130]. In case of anomalous recordings, Langman offered the woman a laparotomy to confirm his suspicions. The technique proved astonishingly effective, since out of the 102 cases in which a significant shift in voltage recording was observed, 95 were confirmed to have malignancies [129,130]. While the exact malignancy location was variable form one patient to another, the cancers were often discovered before the patient had experienced suspicious symptoms. On the whole, the results from these studies led Dr. Burr to hypothesize the existence of “Fields of Life, or L-Fields”.

### 3.2. A Visionary Perspective Awaiting Future Developments

In addition to reporting further citations of studies from the monumental scientific production of Dr. Burr, we believe that some excerpts taken from his “Blueprint for Immortality, The Electric Patterns of Life”, first published in 1972 [131], will help us to appreciate his visionary and pioneering contribution to the field of bioelectricity and, more generally, to the advancement of science and the chance of novel cures for suffering people.

“Electro-dynamic fields are invisible and intangible; and it is hard to visualize them. But a crude analogy may help to show what the fields of life—L-fields for short—do and why they are so important. Most people who have taken high school science will remember that if iron filings are scattered on a card held over a magnet, they will arrange themselves in the pattern of the ‘lines of force’ of the magnet’s field. And if the filings are thrown away and fresh ones scattered on the card, the new filings will assume the same pattern as the old. Something like this happens in the human body. Its molecules and cells are constantly being torn apart and rebuilt with fresh material from the food we eat. But, thanks to the controlling L-fields, the new molecules and cells are rebuilt as before and arrange themselves in the same pattern as the old ones” [131].

Burr had a clear vision of the needs for novel tools and strategies to investigate novel fields and falsify or confirm a remarkable paradigm shift, as the chance of reinterpreting cellular and organ dynamics through the hypothesis that field physics, more than particle physics, may help with navigating the largely unexplored territory of morphogenetic signaling under normal and diseased states.

(“Until modern instruments revealed the existence of the controlling L-fields, biologists were at a loss to explain how our bodies ‘keep in shape’ through ceaseless metabolism and changes of material. Now the mystery has been solved: the electro-dynamic field of the body serves as a matrix or mould which preserves the ‘shape’ or arrangement of any material poured into it, however often the material may be changed” [131]).

The feelings of Dr. Burr, while discovering this new landscape, are well expressed in other excerpts from his work:

(“In the growth and development of every living system there is obviously some kind of control of the processes. As a distinguished zoologist once said, “The growth and development of any living system would appear to be controlled by someone sitting ‘on the organism’ and directing its whole living process. The Field theory suggested that it should be possible to determine the polarity and direction of the flow of energy transformations in the living system. The organism, as a whole, depends on such directives for its continued existence; so also does atypical growth” [131]).

Burr had also provocatively hypothesized that living organisms possessed a global bioelectric field orchestrating and/or emerging from more localized fields and acting as a sort of electrodynamic representation of smaller-scale components (organs, tissues, and cells) of the whole body itself, thus setting the basis for future studies aiming at verifying or falsifying such a hypothesis. (“We had reason to believe that the electro-dynamic field could serve as a signpost for a variety of conditions because our experiments had confirmed our basic assumption. This was that the organism possesses a field as a whole, which embraces subsidiary or local fields, representing the organism’s component parts. We assumed, then, that variations in the subsidiary fields would be reflected in variations in the flow of energy in the whole system—as we had found in ovulation and malignancy” [131]).

A major outcome from Dr. Burr’s work was laying the basis for a transdisciplinary effort in science, fostering the needs for cooperation among committed “researchers” in apparently different disciplines, such as the arts, philosophy, and religion, always aiming at taking a glimpse from the merging of different viewpoints.

This transdisciplinary endeavor is now being propelled by the merging of cellular and developmental biology studies with the most advanced applications in computer science and artificial intelligence (AI), coupled with the development and availability of novel probes designed for the 3D imaging of electric microcurrents at both the single-cell and tissue levels.

### 3.3. Novel Evidence Supporting a Morphogenetic Code

Addressing the intracellular electric fluxes by the aid of a patch or voltage clamp merely allows for the assessment of electric dynamics at the cellular membrane level, which only accounts for about 0.1% of the cell volume. The availability of E-PEBBLEs (photonic explorer for biomedical use with biologically localized embedding), nanosized voltmeters that can diffuse throughout the entire cell volume, is currently providing a novel landscape based upon the 3D profiling of electric fields in living cells and tissues [132,133]. E-PEBBLEs encompass *di-4-ANEPPS*, a fast-responding, *ratiometric*, voltage-sensitive probe, revealing electric field changes as a shift in its fluorescence spectrum emission [132,133]. Notably, E-PEBBLEs revealed the existence of intracellular electric fields other than those traversing the cell membrane, but rather originating inside cells and exhibiting diffusing characteristics through the cytosol and beyond the cellular boundaries [132,133]. These observations are in agreement with the findings discussed herein, indicating that microtubules and microfilaments behave as electrically charged, oscillating circuitries amenable for both intra- and intercellular connectivity. A significant boost in the analysis of bioelectricity, conceived as cell processes involving ions or ion fluxes, has been afforded by the synthesis and availability of fluorescence voltage reporters, including DiBAC4 and CC2-DMPE [134,135]. These dyes, differently from classical electrode-based electrophysiological tools that are constrained to single-cell measurements, can be used in cultured cells, monitoring multicellular areas and volumes, with the chance for monitoring mobile targets and performing measurements over long periods of time. A detailed comprehensive description of the use and characteristics of fluorescent voltage reporters is available elsewhere [134,135]. The use of DiBAC4 and CC2-DMPE, in combination with confocal microscopy analysis, has clearly shown that: (i) membrane potential controls adipogenic and osteogenic differentiation of mesenchymal stem cells [136,137]; (ii) developing neurons form transient nanotubes facilitating electrical coupling and calcium signaling with distant astrocytes [138,139], affording neural-to-glial communication and developmental signaling through a physical form of bioelectronic circuitry; (iii) bioelectric signaling via potassium channels operates as a crucial mechanism in craniofacial patterning [140]; and (iv) altered ion fluxes hamper skeletal morphogenesis, as occurs in defined channelopathies [141].

Taken together, these findings corroborate the notion of a bioelectric memory, modeling the onset of shapes and functions in amphibian embryos and mammalian cells [142,143], prompting consideration for the capability of endogenous bioelectrical networks, such as those associated with microtubular proteins, to store non-genetic patterning information during development and regeneration [144,145].

A major perspective is now emerging from these studies—that endogenous voltage potentials and the microenvironment entail bioelectric signals whose complexity may span from revealing, inducing, and even normalizing cancer [146,147], with the perspective that bioelectric signaling may be part of reprogrammable circuits underlying embryogenesis, regeneration, and cancer [65,148,149,150].

## 4. Conclusions and Future Directions

Deciphering the mechanisms through which physical signaling acts on cell biology, imprinting molecular patterning that progress from the subcellular/cellular level up to the definition of complex shapes and large-scale anatomy, is becoming an increasing effort among scientists. Although this journey started many years ago, only recently has the merging of multiple disciplines in synergistic pathways brought together molecular biologists, physicists, and electronic and software engineers to understand how physical energies may be conveyed to (stem)cells in vitro, as well as in in vivo settings, to afford remarkable modulation of cell behavior, as well as tissue patterning and rescue. Thus, the chance of exploiting the diffusive features of physical energies to modulate and control cell biology in situ is opening the perspective of reprogramming tissue-resident stem and somatic cells, to afford a reverse remodeling of injured tissues without the need for cell or tissue transplantation. In our opinion, significant progress along this line of study may be achieved by the combinatorial use of sophisticated technologies, including AFM, STM, and HSI, coupled with confocal microscopy analyses, to decipher the defined vibrational signatures resulting from the merging of the nanomechanical and electric/electromagnetic levels of cellular activities. We are currently working to extract specific signaling information from these types of recordings, then modulating targeted (stem) cell dynamics through the release of the acquired mechanical and electromagnetic (including light) signals by the aid of ad hoc-designed multifrequency transducers. This is a complex and difficult task, requiring a transdisciplinary approach and the convergence of scientists with different backgrounds and competences. We hope that these efforts will boost the chance among the scientific community of using mechanical vibrations, bioelectricity, and electromagnetic radiation (including light) to develop innovative approaches in the control of epigenetics, tissue morphogenesis, and regeneration, even paving the way for offering additional tools in the handling of oncogenesis and metastatic spreading.

The progressive growth of a transdisciplinary endeavor and of cooperation within a new generation of scientists, as well as the availability of targeted funding programs, may hopefully provide further unfolding in this field.

## Figures and Tables

**Figure 1 ijms-23-03157-f001:**
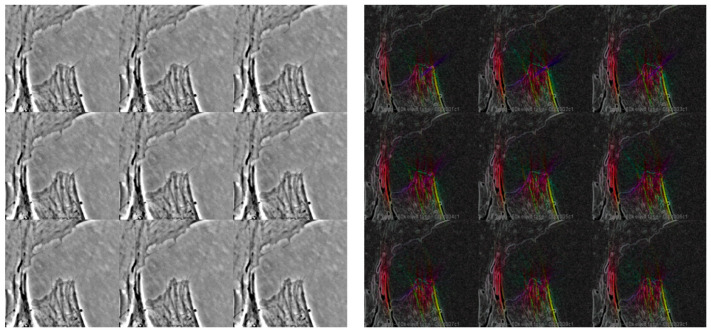
(**Left panel**): time lapse acquisition of cell movement during 3 min. (**Right panel**): directionality map showing vectors, represented as angle and module, of cellular movement of the same frames. The intensity of the movement and the distance covered during a 0.05-Hz acquisition together imply an actin and tubulin turnover rate with higher orders of magnitude of frequencies because of the dimensions of the single monomers that compose the mesh of the cytoskeleton. The microphotographs show a 60× phase contrast acquisition of dermal fibroblast cells.

**Figure 2 ijms-23-03157-f002:**
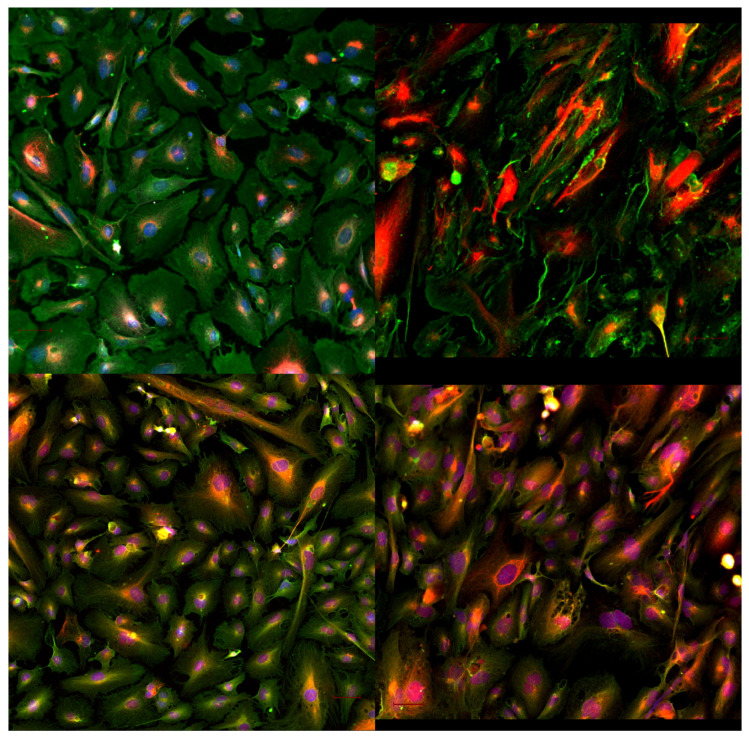
(**Upper panel**): acetylated (red) and polyglutamylated (green) tubulin of control (left) and stretched (right) cardiac fibroblast cells. (**Lower panel**): detyrosinated (red) and tyrosinated (green) tubulin of control (**left**) and stretched (**right**) cardiac fibroblast cells. The rearrangement of the tubulin mesh caused by a mechanical strain not only influences the shape and morphology of the cells but requires an incredible number of finely tuned modifications of the single units that compose the microtubuli in a matter of minutes, all shared and synched by each of the cells that compose the tissue. The microphotographs show a 20× fluorescence acquisition of cardiac fibroblast cells.

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
