# Peer review of "Cell Responsiveness to Physical Energies: Paving the Way to Decipher a Morphogenetic Code"

_ijms, 2022, doi:10.3390/ijms23063157_

Round 1
Reviewer 1 Report
This is an exciting manuscript with Prof. Ventura as a particularly well cited author amongst others. The 21st century is one where complexity, emergent phenomena and self organization as a key scientific challenges. This manuscript poses challenges to the limits of biology as a reductionist endeavor. There are indeed many examples where complexity and complex interactions can play a defining role. Mechanobiology and the role of mechanical motion, even on the nanometer scale, have been well documented and mechanobiology has been shown to show signatures of metastasis including humans. This work provides rich examples to back up their proposals.
The authors propose areas of medical treatment that have a potential to benefit from transdisciplinary research into this emerging field. The use of electromagnetic, bioelectric, mechano sensing and stimulation seems to be a natural development from increased technological availability.
Author Response
We thank this Reviewer for the full appreciation of our work
Reviewer 2 Report
This paper is an enjoyable account of many diverse pieces of biophysics. It is mostly suitable to publish as-is. But, I can suggest one thing to improve value for readers. It is missing something sharp and specific, as a "take-home" message for readers: at least one (or ideally more) very clear examples of the form:
- cells/tissues/organs do amazing phenomenon X that is not yet understood
- we suggest a specific model of how physical energies enable X
- to test this suggestion, experiments Y and Z can be done, and if the result is ABC, then we know the model is falsified (or supported).
As it is, there are a lot of talk about specific mechanisms and "biological shapes and functions" but it is hard to pinpoint what exactly the authors are proposing, how to know if it's correct, and what exactly should be done next (implications or specific directions revealed by this way of thinking). I am 100% on board with the idea that physical energies are the way to understand growth and form, but I was missing here a much more specific statement of what exactly the authors are proposing. For example, "acting as a sort of control software for the molecular patterning" (Abstract) sounds great, maybe tell the reader what specifically is being suggested (make the analogy more explicit - what does it mean that it's software), and how this idea is testable.
Also, I was surprised there's not a single figure in this piece - it would be useful to have a schematic summary of what is discussed, showing some of the vibration machinery they discuss (at the different cells) and how it could work to organize larger-scale biology.
Minor:
- "E-PEBBLEs encompassing DiBAC4 and CC2-DMPE" - I don't believe the experiments described in that paragraph on p. 13 used E-PEBBLES, just the two dyes in solution.
Author Response
“This paper is an enjoyable account of many diverse pieces of biophysics. It is mostly suitable to publish as-is. But, I can suggest one thing to improve value for readers. It is missing something sharp and specific, as a "take-home" message for readers”.
We thank this Reviewer for her/his nice comments and appreciation. In order to comply with the requests of another Reviewer (Reviewer #3), we have substantially re-written several parts of the article, including more mechanistic and precise insights about the scientific findings reported, thus providing a more detailed and interconnected narrative with the main take-home messages throughout the entire manuscript. In the last part, the Conclusion Section now also includes a Future Directions part, where we report the general features of our proposed approaches.
Minor:
- "E-PEBBLEs encompassing DiBAC4 and CC2-DMPE" - I don't believe the experiments described in that paragraph on p. 13 used E-PEBBLES, just the two dyes in solution”.
We thank this Reviewer for noticing this incongruence. We have corrected the text, and the recall to the related references accordingly, also including more detailed information on the use and characteristics of these fluorescent voltage reporters (page 17 of the version with revision tracking).
Reviewer 3 Report
In this review, Tassinari et al. explained the role of physical energies in biological systems based on fundamental models in physics and chemistry. The authors provided an insightful review of chemical signaling and bioelectricity in the context of biomolecular recognition, stem cell biology and morphogenetic field.
This review encompasses a comprehensive compendium of biological phenomena and introduces several conceptual views that are both fundamental and visionary. In addition, the paper is also nicely written. However, the scientific rigor of the manuscript could be improved based on my following comments.
- Concrete scientific findings should be discussed instead of including unnecessary information. For example, in the second paragraph of 3.1, the authors should explain the seminal discoveries by Dr. Durr for his work on electrodynamic fields for living organisms in detail rather than mention ‘eclectic personality’ or ‘his studies were published in extremely relevant journals, including the PNAS and Science.’ Currently, the authors only say that these findings are related to electric stimuli in slime mold and bioelectricity in human ovulation. Detailed scientific findings beyond the simple scope of these studies should be discussed.
- The authors have proposed several novel ideas to perceive biological events. However, more literature work should be included to back up these arguments. For instance, the authors claim that ‘signaling peptides can be regarded as a multitude of oscillatory devices using molecular machines...' This statement will be more convincing by including experimental results from others that support this idea. Similarly, the concepts proposed in the 1st paragraph of 2.4 can be strengthened by including scientific reports that either provide evidence for these claims or can be explained using the proposed analogies.
- Although I appreciate that the authors elaborate on several important biological phenomena with detailed descriptions and discussion, the manuscript is a bit verbose in general. My suggestion is to keep solid scientific reports as the main body of the manuscript and reduce the frequency of generalization/analogy. For example, the paragraph starting with the sentence ‘let’s think of the microtubule like an elastic matrix of nanowires…’ contains a discussion/analogy about an acrobat. Concepts like microtubule should be explained in a more scientific context. In fact, a paper written by most authors of this paper is the one of the best examples to follow: Facchin F, Bianconi E, Canaider S, Basoli V, Biava PM, Ventura C. Tissue Regeneration without Stem Cell Transplantation: Self-Healing Potential from Ancestral Chemistry and Physical Energies. Stem Cells Int. 2018 Jul 3;2018:7412035. doi: 10.1155/2018/7412035. PMID: 30057626; PMCID: PMC6051063.
- Occasional use of direct quotations is acceptable, since I understand that this review is tailored to be visionary. However, extensive use of quotations without paraphrasing or concise summaries, such as in section 3.2, should be avoided.
- Personally, I think the overall writing of this manuscript can be more formal. For instance, expressions like ‘lets’, ‘they’re’, ‘i.e.’ can be reworded. Also, interrogative sentences could be used less often. Examples include the 1st paragraph of the section 2.2 and the 3rd paragraph on page 10.
- Certain words do not need to be capitalized: chemistry, physics, lysine, glutamate, kinome, etc.
Author Response
We thank this Reviewer for the valuable comments. Accordingly, several parts of our revised manuscript have been re-written and expanded to incorporate all her/his criticisms and suggestions.
In particular,
- We have provided and discussed detailed scientific findings beyond the simple scope of the reported studies: 2. From the single-molecule level to the collective behavior: approaching biomolecular recognition, pages 4-6 have been completely re-written, providing and discussing novel mechanistic insights about the mechanical and electromagnetic properties of microtubuli and cytoskeleton, even highlighting some relevant biomedical implications.
- Experimental evidence has been added and discussed, indicating that signaling molecules may not only exploit their individual function but even recognize each other and establish informational networks through specific vibrational patterning (pages 7-9 in the version with revision tracking).
- 4. From the cellular to the tissue level, incorporates more detailed discussion (page 11 in the version with revision tracking).
- Each of the repported studies of Dr. Burr have now been accurately discussed, describing in detail the specific findings and their relevance (pages 14-16 in the version with revision tracking).
- Overall, all concepts and take-home messages have been explained through and within their scientific context, only providing limited analogy afterwards for the sake of clarity.
- Direct quotations are now occasionally used, being also associated with paraphrasing and concise summaries (pages 14-16 in the version with revision tracking).
- Overall, the writing of the manuscript has been kept more formal, avoiding the use of capitalized letters for certain words (chemistry, physics, lysine, glutamate, kinome).
Round 2
Reviewer 3 Report
The authors have satisfactorily addressed most of my concerns. I hereby recommend acceptance in present form.